



# Inclusion of the subgrid wake effect between turbines in the wind farm parameterization of WRF

Wei Liu[1,2], Xuefeng Yang[1], Shengli Chen[1], Shaokun Deng[1], Peining Yu[3], Jiuxing Xing[1]

[1]Institute for Ocean Engineering, Shenzhen International Graduate School, Tsinghua University, Shenzhen, 518055, China

[2]Mingyang Smart Energy Group Corporation, Zhongshan, 528437, China

[3]Shenzhen Institute of Information Technology, Shenzhen, 518172, China

*Correspondence to*: Shengli Chen (shenglichen@sz.tsinghua.edu.cn), Peining Yu (peining.yu@sziit.edu.cn)

**Abstract.** Wind farms, as an important renewable energy source to combat climate change, have had explosive development in recent years. Assessing impacts of wind farms on atmospheric and marine environments requires an accurate parameterization of wind farms in atmospheric models. The current wind farm parameterization scheme (Fitch et al. 2012) in WRF plays an important role in the study of impacts of wind farms. The scheme, however, has some shortfalls, e.g., does not consider the wind wake behind turbines inside a grid cell. In this research, the Fitch scheme in WRF is modified by inclusion of the wake effect of wind turbines. Based on an engineering wake model of a turbine, a wake superposition coefficient and an angle correction coefficient are proposed. A solution model for the inflow wind speed is established to obtain the angle correction coefficient. Other coefficients in the engineering wake model are calculated based on the CFD results. These coefficients are added in the WRF to improve the wind farm parameterization, and sensitivity experiments are conducted. Model results show that the new improved scheme significantly increases wind energy, output power and turbulent kinetic energy in the wind farm area compared with the original scheme. Sensitivity experiments also reveal that, with enlarged model grid size and shortened turbine spacing, the subgrid wake effect becomes more significant, and the new scheme shows more advantages.

Keywords: wind farm parameterization; wake effect; WRF



## 1 Introduction

Wind power, as a pollution-free, renewable and widely distributed energy, has gradually become the top priority of international energy transformation. In the process of adjustment of energy structure, promotion of energy production and consumption revolution, the development of wind power industry has played a pivotal role. As a strategic emerging industry, wind power has been guided by a large number of industrial policies and driven deeply by market demand, thus achieving a rapid development period. According to the latest Global Wind Power Industry Report 2022 released by the Global Wind Energy Council (GWEC), by 2021, the quantity of world's new wind power installed capacity is up to 93.6 GW, the cumulative installed capacity has reached 837GW, an increase of 12% over the previous year. With the increase of the scale of wind farms and the size of a single turbine, the impacts of wind farms on the surrounding atmospheric environment is also enlarging. Fitch et al. (2012) found that the wind speed decay in the offshore wind farm could reach 16%, and could extend to the downstream area of 60 km. Christiansen et al. (2006) used the SAR radar data to study the impact of two large offshore wind farms in Denmark, and reported that the average wind speed was reduced by 8%~9%, and the wind speed decay zone could extend to 5~20 km along the wind direction. Roy and Traiteur (2010) showed that the vertical mixing is enhanced due to the turbine wake effect during operation. In the stable atmosphere at night, the vertical mixing is enhanced due the turbine wake effect, which leads to the near-surface's warming, while in the unstable atmosphere during the day, the near-surface is cooling. Fiedler et al. (2011) argued that the precipitation in the southeast of and around the wind farms in multiple states is reduced by 1% due to the wind farms, and Vautard et al. (2014) showed that the change of winter precipitation can be 0~5%. Barrie and Kirk-Davidoff (2010) simulated effects of large-scale onshore wind farm and found that the operation of large scale wind farm would change the surface roughness and promote the generation of cyclones, thus causing atmospheric disturbance and significantly affecting atmospheric circulation. In short, the construction of large scale wind farm has a certain impact on climate factors at both global and local scales.

At present, numerical model simulations are the mainly method for researching the environmental impact of wind farms. Among these numerical models, Weather Research & Forecasting (WRF), developed in collaboration with the National Center for Atmospheric Research (NCAR) and others, plays an important role in this scope and has been widely accepted by researchers. The grid resolution used in WRF is generally greater than 1 km, and the size of the turbine is only a few hundred meters, numerical





models, therefore, cannot resolve wind turbines and the effect of wind farms directly, and it is necessary
to use parameterization to characterize the effect of the wind turbine on the atmosphere. The
parameterization scheme applied in WRF is proposed by Fitch et al. (2013) which describes wind farms
as momentum sink and turbulent source in the environment. Many scholars have employed this scheme
to study the environmental impact of wind farms (Boezio and Ortelli, 2019; Jacondino et al., 2021; Pryor
et al., 2019). However, the scheme ignores the influence of the wake of the front turbine on the rear
turbine, which causes obvious errors. With the rising scale of wind farm in future, the subgrid wake effect
will be more prominent. Therefore, it is of great significance to explore the correction method of the
subgrid wake effect in WRF wind farm parameterization to improve the accuracy of wind farm
representation.
For the subgrid wake effect, previous researchers have proposed some solutions. Abkar and Port´e-Agel
(2015) tried to average the simulation results of LES and obtained a correction coefficient $\xi$ to correct
the error of subgrid wake effect, but there is no universal prediction model or function for the correction
coefficient $\xi$. Pan and Archer et al. (2018) combined the simulation results of LES with the relevant
geometric parameters of wind farm layout, and proposed a "hybrid parameterization" scheme, and
experimental results show that the hybrid parameterization scheme also has a good effect on the
correction of subgrid wake effect. These above schemes use LES technology to achieve the purpose of
correcting subgrid wake effect. However, LES requires a large amount of computation, which limits the
usage of this method. Even if the LES simulation of wind farm is processed by the actuator model, it will
still consume huge computing resources in the face of the tendency of ultra-large wind farm construction
and the substantial increase in the number of turbines in a single wind farm. Elshafei et al. (2021) used
the spatiotemporal fusion data of deep multi-fidelity Gaussian regression and nonlinear autoregressive
algorithms to combine the simulation output of WRF with the field observation data to improve the
simulation accuracy. However, the observational data used is difficult to obtain.
Although the development of wind farm parameterizations in WRF has undergone several revision
iterations, there is still significant room for improvement in the handling of subgrid wake in this model.
This study attempts to correct subgrid wake effect errors in a new way, namely, through a simple
engineering wake model. Sect. 2 of the paper introduces the parameterization principle of wind farms in
WRF and a correction principle of subgrid wake effect based on engineering wake model. Sect. 3 displays
the correction and calibration results of the engineering wake model. In Sect. 4, effects of the proposed
new parametric scheme are analyzed and a series of sensitivity experiments are carried out.



## 2 Principle of subgrid wake effect correction

### 2.1 Wind farm parameterization in WRF

The WRF model is a completely compressible and non-hydrostatic multi-layer forecasting model for small and medium scale weather systems, developed by the National Oceanic and Atmospheric Administration, the National Center for Atmospheric Research and other agencies. The horizontal resolution of the grid in WRF model generally ranges from 1 to 10 km, which is larger than the feature scale of some motion elements. In order to better describe physical processes of these subgrid scale motions, it is necessary to use parametrization methods for representing the interaction between the solvable scale and the unsolvable scale. WRF model utilizes parameterization schemes of physical processes including short-wave radiation and atmospheric long-wave radiation, microphysical processes, boundary layer, cumulus convection, etc., to improve simulation accuracy (Skamarock et al, 2008). Since the height of wind farms is in the order of 100 meters which are located in the atmospheric boundary layer, it is necessary to supplement the boundary layer parameterization scheme when exploring the impact of wind farms on the environment with WRF. According to the "momentum sink - turbulent source" theory proposed by Fitch et al. (2013), parameterization of wind farms is realized by adding a momentum trend term Eq. (1) to the momentum equation and a turbulent energy generation term Eq. (2) to the equations of turbulent energy. In addition, a power generation term Eq. (3) is introduced to calculate the power output of the entire wind farm.

$$\frac{\partial V_{ijk}}{\partial t} = -\frac{1}{2} \frac{N_t^{ijk} C_T(V_{ijk}) V_{ijk}^2 A_{ijk}}{(z_{k+1} - z_k)}, \tag{1}$$

$$\frac{\partial TKE_{ijk}}{\partial t} = \frac{1}{2} \frac{N_t^{ijk} C_{TKE}(V_{ijk}) V_{ijk}^3 A_{ijk}}{(z_{k+1} - z_k)}, \tag{2}$$

$$\frac{\partial P_{ijk}}{\partial t} = \frac{1}{2} \frac{N_t^{ijk} C_P(V_{ijk}) V_{ijk}^3 A_{ijk}}{(z_{k+1} - z_k)}, \tag{3}$$

where $V_{ijk}$ is the wind vector at the grid ($i, j, k$); $TKE_{ijk}$ is the turbulent kinetic energy at the grid ($i, j, k$); $P_{ijk}$ for the power output at the grid ($i, j, k$); $C_T(V_{ijk})$, $C_{TKE}(V_{ijk})$, $C_P(V_{ijk})$ are turbine thrust coefficient, turbulent kinetic energy generated coefficient and power factor, respectively, which are functions of velocity; $A_{ijk}$ is turbine swept area; $N_t^{ijk}$ is the number of the turbines. These equations show that the inflow wind speeds of all turbines are the same within one grid in this original parameterization. At the same time, it can be seen that the core variable is the inflow wind speed of the



turbine which determines the accuracy of the parameterized scheme. Therefore, this study starts with the
correction of the inflow wind speed of the turbine by which the error of the subgrid wake effect based on
the engineering wake model can be reduced, so as to improve the accuracy of the simulation of wind
farm effect in WRF.
In this study, a wake superposition coefficient and angle correction coefficient will be used to correct the
inflow wind speed of the turbines, and the specific relationship is shown in Eq. (4). The wake
superposition coefficient considers the wind speed change due to the wake superposition in front of each
turbine when the inflow wind speed is perpendicular to the wind farm, and the angle correction
coefficient is further corrected for any wind direction. In fact, there is a wind direction angle between
inflow wind and wind farm layout. The wake superposition coefficient corrects the inflow wind speed
under the condition of 0 ° wind direction angle, while an angle correction coefficient is to correct the
inflow wind speed under any θ ° wind direction.
$$u = C_a \cdot C_b \cdot u_0,\qquad(4)$$
where $C_a$ is the wake superposition coefficient, $C_b$ indicates the angle correction coefficient, and $u_0$
denotes the original wind speed.
**2.2 Wake superposition coefficient**
The wake superposition coefficient is proposed based on the wake analytic model and the wake
superposition model. The analytical model of a single turbine wake is a mathematical expression of the
distribution of turbine wake velocity. Currently, there are many expressions of wake analytic models,
among which the Jensen model (Jensen et al., 1984) appears earlier and is widely used.

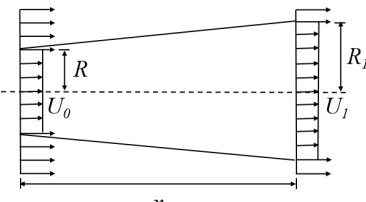


Figure 1: Schematic diagram of the wake analytic model.
This model assumes that the expansion of wake width behind the turbine is linear along the flow direction



(Fig. 1), and that the loss of wake speed is related to $C_T$. According to the one-dimensional momentum
theorem and the mass conservation theorem, the relation between the wind speed $u_i$ and the inflow speed
$u_0$ can be given as (Jensen et al., 1984):
$$u_i = u_0[1 - (1 - \sqrt{1 - C_T})(\frac{R}{R+kx})^2],$$ (5)
where $u_0$ represents the inflow wind speed of turbine $i$, $u_i$ represents the wake speed of the turbine $i$ at the
downstream position $x$; $C_T$ is the thrust coefficient of the turbine; $R$ is the radius of the turbine's sweeping
area; and $k$ is the wake expansion coefficient, which is related to the roughness coefficient of the ground.
Eq. (5) only describes the speed distribution of the wake behind a single turbine. In practice, the
downstream turbine is often partly blocked by the upstream turbine, therefore it is necessary to introduce
a wake superposition model. The wake superposition models proposed mainly include the quadratic sum
superposition model, the linear superposition model and the energy conservation superposition model.

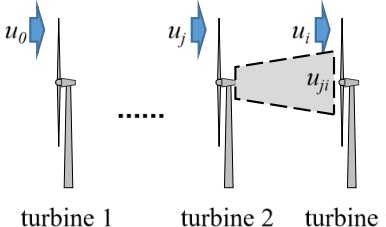


Figure 2: Schematic diagram of quadratic sum wake superposition model.
Among them, the more widely used is the quadratic sum superposition model proposed by Katic (1986)
(Fig. 2), and is expressed as:
$$(1 - \frac{u_i}{u_0})^2 = \sum_{j=1}^{n}(1 - \frac{u_{ji}}{u_j})^2,$$ (6)
where $u_i$ represents the inflow wind speed of turbine $i$; $u_0$ is the initial wind speed before superposition;
$n$ is the total number of superposition turbines in front of turbine $i$; $u_j$ denotes the inflow wind speed of
turbine $j$; and $u_{ji}$ represents the wake wind speed of turbine $j$ at turbine $i$. According to Eq. (2.5), the
expression of $u_{ji}$ is:
$$u_{ji} = u_j[1 - (1 - \sqrt{1 - C_T})(\frac{R}{R+kx_j})^2],$$ (7)
where $x_j$ represents the distance between turbine $j$ and turbine $i$. Substituting Eq. (7) into Eq. (6):





$$u_i = u_0[1 - \sqrt{\sum_{j=1}^{n}[(1 - \sqrt{1 - C_{Tj}})(\frac{R}{R+kx_j})^2]^2}], \tag{8}$$
By contrast with Eq. (4), the expression of wake superposition coefficient can be given by:
$$C_a = [1 - \sqrt{\sum_{j=1}^{n}[(1 - \sqrt{1 - C_{Tj}})(\frac{R}{R+kx_j})^2]^2}], \tag{9}$$
**2.3 Angle correction coefficient**
The same as the wake superposition coefficient, because the turbine effect in the grid is directly
superimposed, it is only necessary to correct the total effect of the turbine wake superposition effect
under the condition of θ ° wind direction angle. The total effect of the turbine's wake superposition is
averaged for each turbine, and the angle correction coefficient is used to express the effect of the averaged
wind speed of a single turbine on the wake superposition. That is to say, instead of correcting the specified
effect of a single turbine, only the overall deviation of the wind farm relative to the 0 ° wind direction is
evaluated. For additive terms Eqs. (1-3), the velocity variables have different powers, so the angle
correction coefficient needs to be divided into quadratic and cubic according to the power number of the
additive terms.
According to Eq. (1), after correcting using the wake superposition coefficient, that is, when the wind
direction is 0°, the total effect of the turbine in one grid for the momentum trend term is:
$$\frac{\partial V_{ijk}}{\partial t} = -\frac{1}{2}\frac{A_{ijk}\sum_1^n C_T(v_{n0})v_{n0}^2}{(z_{k+1}-z_k)}, \tag{10}$$
where $v_{n0}$ is the inflow wind speed of the turbine in 0 ° wind direction. Introducing the quadratic angle
correction coefficient when the wind direction is θ °, the total effect of the turbine in the momentum trend
term is:
$$\frac{\partial V_{ijk}}{\partial t} = -\frac{1}{2}\frac{A_{ijk}\sum_1^n C_T(C_{b2}\cdot v_{n0})(C_{b2}\cdot v_{n0})^2}{(z_{k+1}-z_k)} = -\frac{1}{2}\frac{A_{ijk}\sum_1^n C_T(v_{n\theta})v_{n\theta}^2}{(z_{k+1}-z_k)}, \tag{11}$$
where $v_{n\theta}$ is inflow wind speed of the turbine in $\theta$ ° wind direction. Therefore, the quadratic angle
correction coefficient $C_{b2}$ is expressed as:
$$C_{b2} = \sqrt{\sum_1^n v_{n\theta}^2/\sum_1^n v_{n0}^2}, \tag{12}$$
Similarly, from the equation of turbulent kinetic energy Eq. (2), the cubic angle correction coefficient
$C_{b3}$ is given by:





$C_{b3} = \sqrt[3]{\sum_1^n v_{n\theta}^3 / \sum_1^n v_{n0}^3}$,                                    (13)
The reason why two correction coefficients are computed in two steps, instead of directly calculating the
inflow wind speed of a single turbine in one step, is that the usage of two correction coefficients will
reduce much computing consumption. As the scale of wind farm tends to be larger in future, the
consumption of computing resources in the simulation process will increase greatly, therefore it is of
great application significance to simplify the calculation.
The calculation of $v_{n\theta}$ and $v_{n0}$ in Eq. (12) and (13) is carried out through the wind farm modeling. As
shown in Fig. 3, in the modeling process, for a wind farm composed of $n$ turbines, $l_n$ is the windward
distance of each turbine along the wind direction angle of $\theta°$, and $n$ turbines are numbered according to
the order of $l_n$ from small to large: $1...i,j...n$, where turbine $i$ is upstream of turbine $j$. The distance
between turbine $i$ and turbine $j$ along the wind direction is denoted $l(i,j)$, and the wake wind speed of
turbine $i$ at turbine $j$ is denoted $v(i,j)$, then $l(i,j)$, $v(i,j)$ can form an $n \times n$ upper triangular matrix, denoted
as the distance matrix $\mathbf{L}$ and the wind speed matrix $\mathbf{V}$ (Eq. 14). The element $v(i,i)$ on the diagonal of $\mathbf{V}$
represents the inflow wind speed of turbine $i$ at a wind direction angle of $\theta°$, i.e. $v_{n\theta}$.

$$V = \begin{bmatrix} v(1,1) & \dots & v(1,i) & \dots & \dots & v(1,n) \\ & \dots & \dots & \dots & \dots & \dots \\ & & v(i,i) & v(i,j) & \dots & \dots \\ & & & v(j,j) & \dots & \dots \\ & & & & \dots & \dots \\ & & & & & v(n,n) \end{bmatrix}$$                    (14)

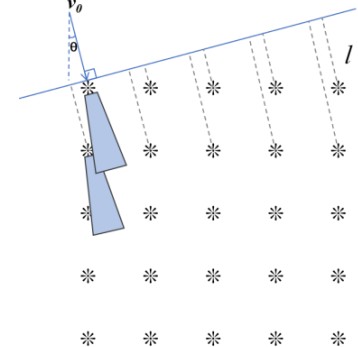


204                          Figure 3: Schematic diagram of wind farm modeling principle.

The wind velocity matrix $\mathbf{V}$ is calculated in the row sequence. According to Jenson wake analytic model
Eq. (5), elements $v(i,j)$ in row $i$ of the wind speed matrix $\mathbf{V}$ can be obtained as:



$v(i,j) = v(i,i)[1 - (1 - \sqrt{1 - C_T})(\frac{R}{R+k\cdot l(i,j)})^2]$,     (15)
Due to the superposition of upstream turbines, the calculation of downstream turbine inflow wind speed
needs to consider the superposition effect of the wake. According to the quadratic sum superposition
model Eq. (6), the $v(i+1,i+1)$ can be calculated as :
$v(i + 1, i + 1) = v(1,1)\left[1 - \sqrt{\sum_{n=1}^{i} \gamma(n, i+1)\left(1 - \frac{v(n,i+1)}{v(1,1)}\right)^2}\right]$,     (16)
Given the inflow wind farm wind speed $v(1,1)$, combining the Eq. (15) and Eq. (16),     all the elements
in the matrix **V** can be solved, namely the inflow wind speed $v_{n\theta}$ of all wind turbines in the wind farm
in $\theta$ ° wind direction angle.
Eq. (6) applies to the situation when the wake is completely overlapping, i.e., the downstream turbine is
completely in the wake of the upstream turbine. In practice, due to the existence of wind direction angle,
the downstream turbine is not completely in the wake of the upstream turbines, hence we introduce a
shielding factor $\gamma(i,j)$ of turbine $i$ to turbine $j$ as in Eq. (16). The shielding factor, representing the degree
to which the upstream turbine's wake affects the downstream turbine, is the proportion of the overlap
area between the upstream turbine's wake and the downstream turbine's disk surface to the swept area of
the downstream turbine's impeller. The distance between turbine $i$ and turbine $j$ perpendicular to the wind
direction is $X(i,j)$, and the wake radius of turbine $i$ at turbine $j$ is $R(i,j)$, then
$X(i,j) = l(i,j) \cdot tan\theta$,     (17)
$R(i,j) = k \cdot l(i,j) + R$,     (18)
where θ is the wind direction angle, $k$ is the coefficient of wake expansion, $R$ is the radius of the impeller's
sweeping area, and $l(i,j)$ is the distance between turbine $i$ and turbine $j$ in the windward direction.

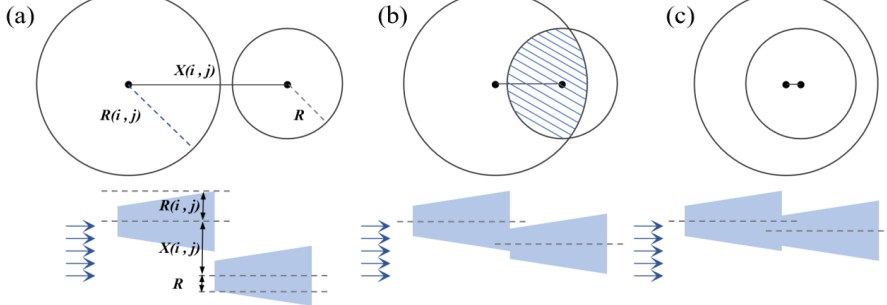


Figure 4: Schematic diagram of the shielding factor calculation(a) no shielding; (b) partial shielding; (c) complete
shielding.



As shown in Fig. 4, the shielding factor has three types of situation:
(1) when $X(i,j) > R(i,j) + R$ (Figure. 2.4a), i.e., the downstream turbine impeller is completely not in the
wake of the upstream turbine, $\gamma(i,j)=0$;
(2) when $R(i,j) - R < X(i,j) < R(i,j) + R$ (Fig. 2.4b), i.e., part of the downstream turbine impeller is in the
wake of the upstream turbine, $\gamma(i,j) = S_{sd} / S_{tb}$;
$$S_{sd} = R(i,j)^2 arccos \frac{R(i,j)^2+X(i,j)^2-R^2}{2R(i,j)X(i,j)} + R^2 arccos \frac{R^2+X(i,j)^2-R(i,j)^2}{2RX(i,j)} -$$
$$X(i,j)R(i,j)sin(arccos \frac{R(i,j)^2+X(i,j)^2-R^2}{2R(i,j)X(i,j)}), \tag{19}$$
$$S_{tb} = \pi \cdot R^2, \tag{20}$$
(3) when $X(i,j) < R(i,j) - R$ (Fig. 2.4c), i.e., the rotating surface of the downstream turbine impeller is
completely in the wake of the upstream turbine, $\gamma(i,j)=1$.
**3 Correction of engineering wake model**
**3.1 CFD experiments of wake**
When calculating the inflow wind speed of the turbine at different wind angles, the wake expansion
coefficient plays an important role. When the total number ($n$) of turbines in a wind farm is 25, the
distance between turbines is 5 times the turbine diameter, and the inflow wind speed $v_0$=10 m/s, the
dependence of the quadratic and cubic angle correction coefficients on the wake expansion coefficient
can be seen in Fig. 5.

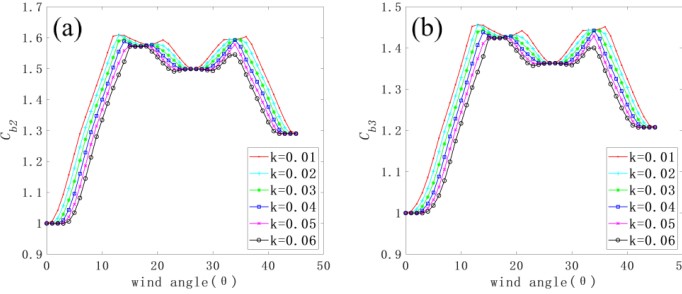


Figure 5: Relation between the angle correction coefficient with the wake expansion coefficient (k) (a) the quadratic
angle correction coefficient; (b) the cubic angle correction coefficient.
With the rising of wake expansion coefficient, both angle correction coefficients diminish. Because the
average inflow wind speed of the turbine at 0 ° wind angle is the same, and with the increase of expansion



coefficient, the distribution of the wake along the downstream gradually becomes divergent. For the case
of θ ° wind angle, the downstream turbine is affected more by the upstream turbine, and the inflow wind
speed becomes smaller, hence the two angle correction coefficients are reduced due to Eqs. (12) and (13).
It is concluded that the change of wake expansion coefficient has a significant influence on the angle
correction coefficient.
The quadratic sum superposition model (Eq. 6) is used in the model for solving the angle correction
coefficient. In the use of quadratic sum superposition model, it is assumed that the inflow wind speed of
the turbine is evenly distributed. However, in the actual situation, the inflow wind speed of the turbine is
affected by ground friction in the form of wind profile distribution. In short, when the wake analytic
model and wake superposition model are used in this study, certain corrections need to be made according
to the CFD simulation results of the turbine's wake.

**3.2 Calibration of the wake expansion coefficient**

The expansion coefficient is examined by the CFD modelling of a single turbine under different inflow
wind speed conditions. In the single turbine experiment, the incoming wind speed of the turbine is in the
form of wind profile. There are 9 groups of experiments are set according to the wind speed at the hub
height, with the wind speed at the hub height ranging from 3 to 19 m/s (an interval of 2 m/s). As shown
in Fig. 6, speed monitoring surfaces are set $1D$ apart in the downstream direction, and two speed
monitoring lines are set in the horizontal and vertical directions on each speed monitoring surface, and
100 monitoring points are set on each monitoring line to monitor the speed amplitude.
After obtaining the wind speed amplitude at the monitoring line in the wake, it is necessary to determine
the boundary of the wake in the horizontal and the vertical direction according to distribution of the speed
amplitude. Then the relationship between wake radius and wake distance can be obtained. With a linear
fit to the change of the wake radius at each inflow wind speed, the slope of the fitted line is the wake
expansion coefficient $k$. After calculation, the relationship between wake expansion coefficient and
inflow wind speed is shown in Fig. 7. The wake expansion coefficient remains relatively stable for the
low wind speed (3-9 m/s), but has a significant linear growth for the high wind speed (9-19 m/s), reaching
to a value of 0.019 for 19 m/s. These wake expansion coefficient is applied to the wake analytical model
(Eq. 9).





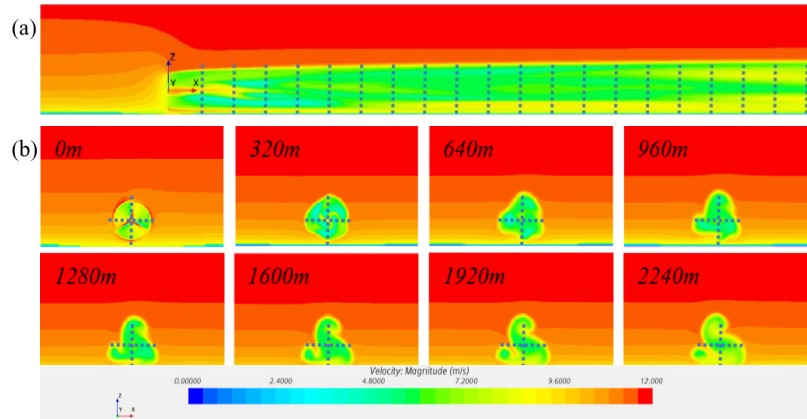


Figure 6: Schematic diagram of monitoring surface setup in the turbine wake (a) Parallel wake profile; (b) Vertical
wake profile.

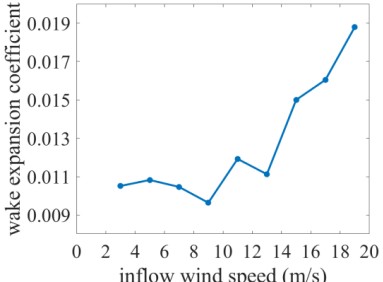


Figure 7: Dependence of wake expansion coefficient k on the inflow wind speed.

### 3.3 Correction of the wake superposition model

The wake superposition model is corrected based on the results of the CFD two-turbine wake experiment.
The experiment settings are shown in Fig. 8. The distance between turbines is 5$D$, and 7 experiments are
carried out within the working wind speed ranging from 7 to 19 m/s with an interval of 2 m/s.

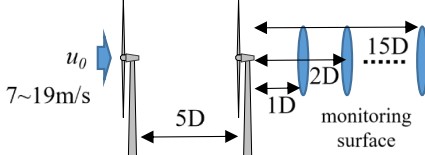


Figure 8: The experiment of two-turbine wake superposition.



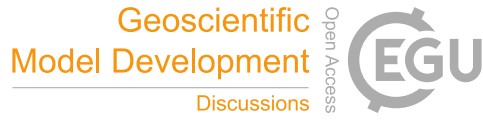

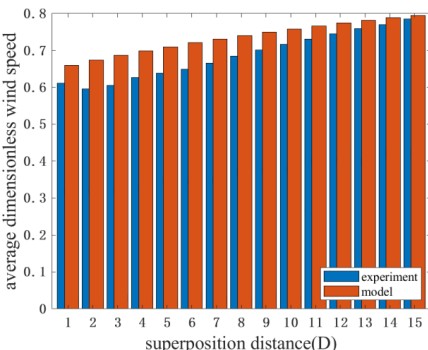


Figure 9: Comparison of the average dimensionless wind speed behind the second turbine between the

superposition experiment and the superposition model. The dimensionless wind speed is obtained by being

dividing by the inflow wind speed.


It can be seen from the comparison results (Fig. 9) that there are significant differences between results
of the CFD experiment and the engineering model. Results calculated using the engineering
superposition model are slightly higher than that simulated using the CFD experiment, and their
difference gradually diminishes with the increase of the distance. Such differences are mainly due to the
assumption that the inflow wind speed is evenly distributed at different altitudes in the calculation of
wake superposition model, while the CFD simulation adopt the inflow wind speed with a wind profile
closer to the actual situation. This assumption is one of the error source of the wake superposition model
in this study, so it is necessary to correct the wake superposition model according to the CFD results to
improve the accuracy of the wind farm model, further improve the accuracy of the angle correction
coefficient, and finally make the correction of the subgrid wake effect more accurate. Thus a correction
factor ($C_{over}$) is proposed as in Eq. 21. The dimensionless velocity of the experiment and the model is
averaged over the wake distance, and the ratio between the experiment simulation ($\bar{v}_{sim}$) and engineering
model ($\overline{v}_{mod}$) is 0.93. By applying this coefficient to Eq. (9), the correction of the wake superposition
model can be completed.
$$C_{over} = \bar{v}_{sim}/\overline{v}_{mod}, \tag{21}$$
**4. Implement of new parameterization and sensitivity experiments**
**4.1 Implement of new parametrization**



The wake superposition coefficient ($C_a$), the quadratic angle correction coefficient ($C_{b2}$) and the cubic
angle correction coefficient($C_{b3}$) are used to modify the original parameterization scheme of wind farm
in WRF. The differences between the original and new schemes are compared, and the correction effect
of the subgrid wake effect is verified and discussed.
In the experiments, the model domain is set from 17.68° N to 27.16° N, from 109.38° E to 122.62° E
(Fig. 10). The wind farm is located in an area between 21.70° N~ 22.50° N, 115.14° E~ 116.00° E . The
type of turbine used in the WRF simulation is the same as that used in the CFD simulation. The turbines
are arranged parallel to the longitude and latitude lines with an equal spacing of $5D$. The total number of
turbines is 25600, and the output power of a single turbine is 3 MW.
The experimental area nests two layers of inner and external meshes with grid resolutions of 2.8 km and
8.4 km, respectively, and 30 layers are used in the vertical direction. The wind farm is located in the inner
area of the nested model. The inner grid spacing is equal to 5 times the turbine layout spacing, i.e., there
are 25 turbines in the one grid cell.
Physical and dynamic schemes used in the simulations are identical to what was performed in the
previous studies (Hong et al.,2006; Mlawer et al.,1997; Fouquart et al.,1991; Grell et al.,2002;
Nakanishim et al.,2009). For the land surface process, the parameterization scheme of heat diffusion is
adopted. Goddard parameterization scheme is used for short-wave radiation process, while RRTM
parameterization scheme is used for atmospheric long-wave radiation process. For microphysical process,
WSM6 parameterization scheme is selected to improve the accuracy of vertical profile and reduce the
influence of time step on physical parameterization scheme. The Grell-Freitas scheme is employed to
parameterize cumulus cloud. MYNN-2.5 is selected for the parameterization scheme of the planetary
boundary layer, which contains various physical processes in detail, and can simulate the influence of
wind farm on atmospheric boundary layer more accurately. The WRF model is initialized at 0000 UTC
on 4 January 2022, using the data from the National Centers for Environmental Prediction Global
Forecast System (GFS). All the simulations are run for about 3 days.
It can be seen that there is a relatively stable northeast wind in the wind farm area at this time (Fig. 10)
and the wind speed is moderate, about 10 m/s. The kind of speed belongs to the critical wind speed of
the turbine to reach the rated power, which is conducive to observe and compare the difference
phenomenon at this moment.

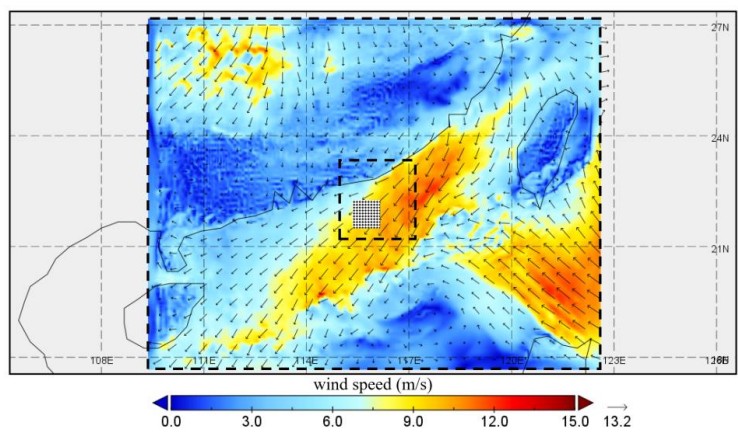


Figure 10: Snapshot of wind vectors and wind speed on 4 January 2022. The dash black lines denote the inner and
external model domains, respectively. The white region with black dots represent the wind farm.

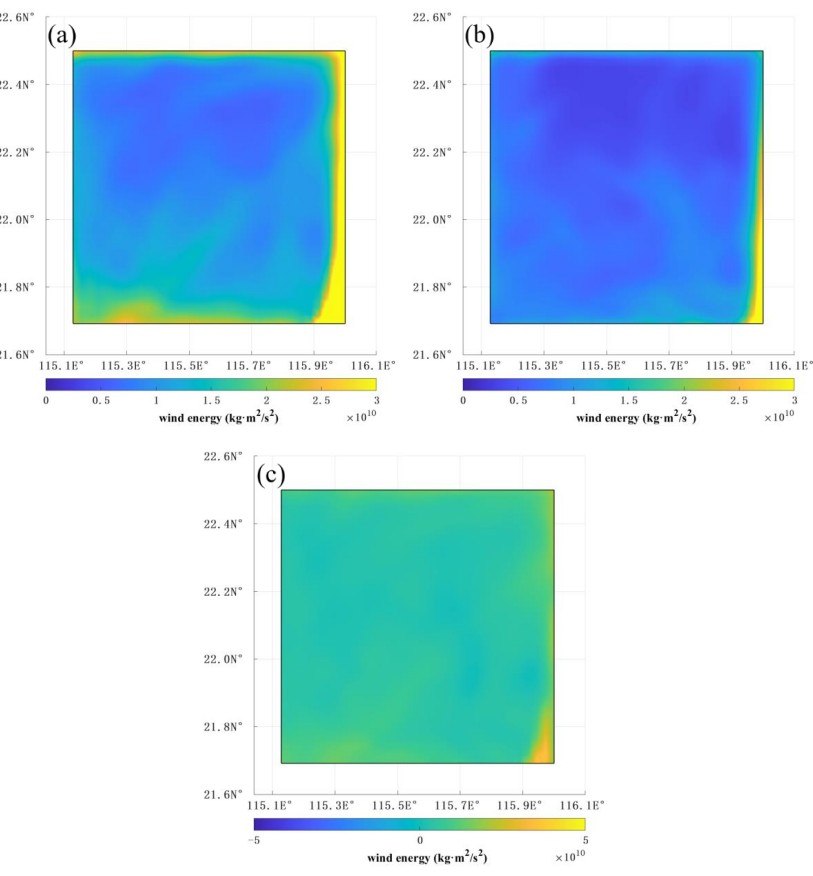


Figure 11: Comparison of wind energy distribution for (a) the new parameterization scheme; (b) the original



347                                  parameterization scheme; (c) their difference.

As shown in Fig. 11, the wind energy inside the wind farm region using the new scheme, which is
$1.44\times1013$ kg·m$^2$/s$^2$ in total, is higher than that of the original scheme ($8.54\times1012$ kg·m$^2$/s$^2$), increasing
by 68%. When the subgrid wake correction is added to the new parameterization scheme of wind farm,
the inflow wind speed of the rear turbine is reduced. Therefore, the absorption of wind energy in the wind
farm region is reduced. At the same time, the total wind energy becomes greater than that in the original
scheme, reducing the error of overestimation of wind speed attenuation in the original scheme. The wind
energy in the upstream edge region of the wind farm is significantly enhanced. This is because in the
new parameterization scheme, the kinetic energy absorption of the grid in the upstream region is reduced
more, so that the wind energy is more fully developed downstream and resulting in greater wind energy
in the edge region of the wind farm.



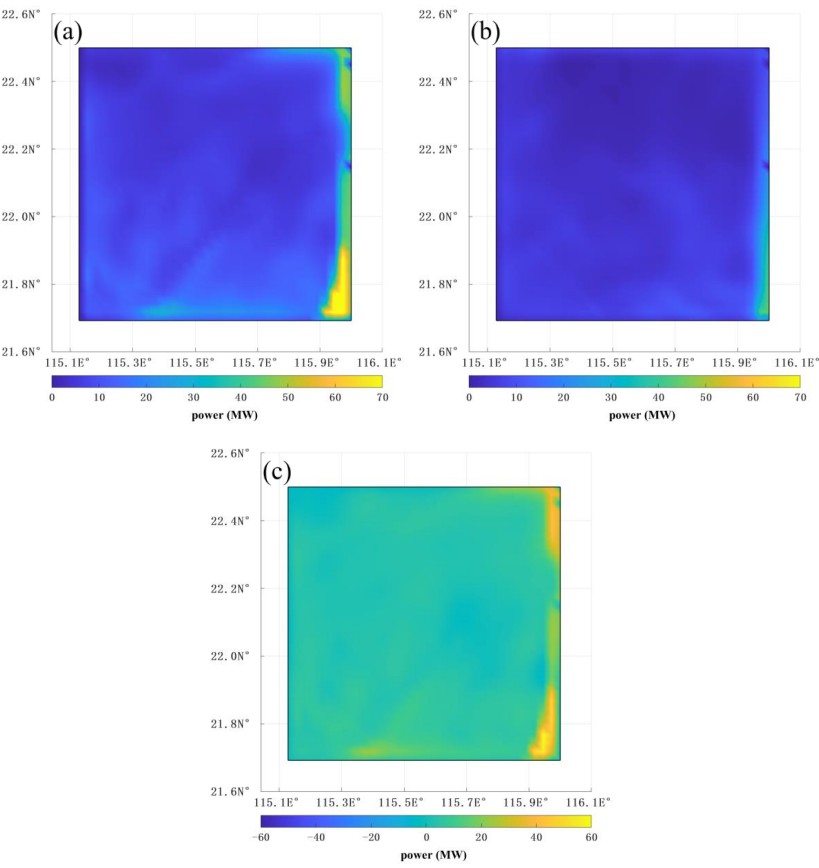


Figure 12: Comparison of power output for (a) the new parameterization scheme; (b) the original parameterization
scheme; (c) their differences.
The power output in the new scheme is higher than that in the original scheme (Fig. 12). The total regional
power output of the wind farm in the new scheme is calculated and the result is 11639.56MW, while that
in the original scheme is 5703.2MW, with an increase of 104%. This is because after the new scheme
corrects the excessive kinetic energy absorption error of the turbine, the wind speed in the wind farm
area increases, thus increasing the instantaneous output power of all turbines. Therefore, the output power
generated by the wind farm increases.



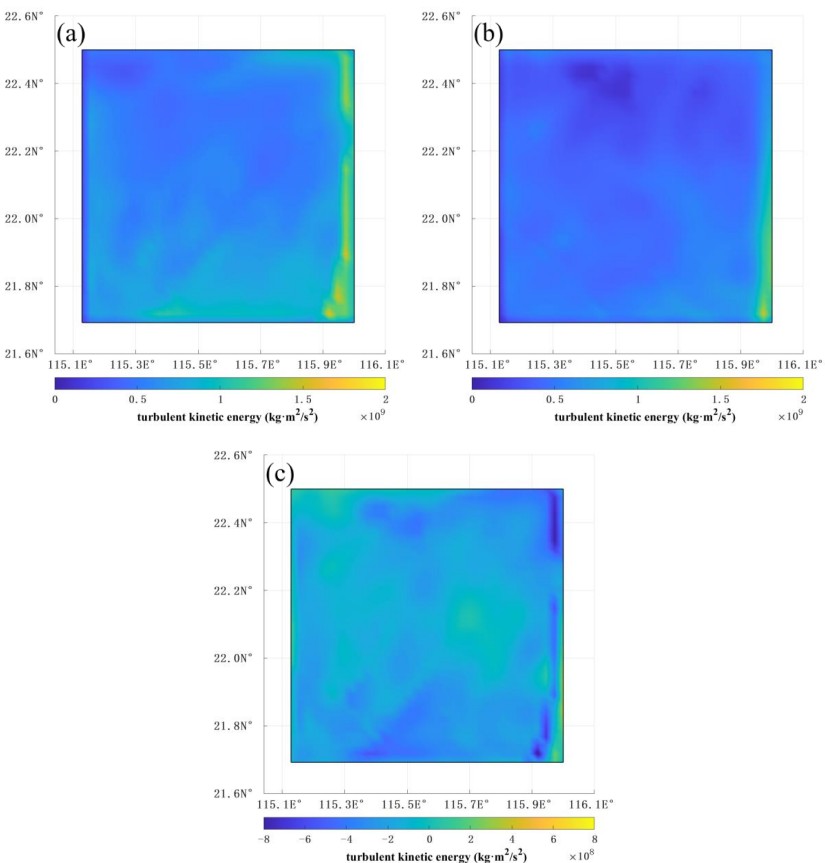


369    Figure 13: Comparison of turbulent kinetic energy for (a) the new parameterization scheme; (b) the original

370          parameterization scheme; (c) their differences.

371  As shown in Fig. 13, the turbulent kinetic energy of the wind farm region in the new scheme is also

372  enlarged. The total regional turbulent kinetic energy under the new scheme is $7.00×1011$ kg·$(m^2/s^2)$, while

373  that in the original scheme is $4.96×1011$ kg·$(m^2/s^2)$, with a rise of 41%. It can be seen from the expressions

374  of the power output and turbulent source terms in the parameterization principle of wind farm Eqs. (2

375  and 3) that power output and turbulent kinetic energy have the same tendency. Therefore, the growth of

376  power output is bound to be accompanied by the enhanced turbulent kinetic energy generation.

377  In summary, compared with the original parameterization scheme, the simulation results of the new

378  scheme have increased significantly for the simulation results of each parameter quantity, and the

379  differences between the two schemes also conform to the relevant law. Therefore, the rationality and



feasibility of the new scheme for the correction of subgrid wake effect can be proved. It forms a
foundation for exploring the further optimization of the new parameterization scheme and its sensitivity
to wind farm parameters.

**4.2 Sensitivity experiments**


A series of model experiments is conducted to investigate the sensitivity of the new scheme to wind farm
parameters. Experiments are divided into two groups: the Grid experiment and the Space experiment. In
all experiments, the simulation area, simulation time and other parameterization schemes are the same
as in the last section. Details of the settings in each group of experiments are shown in Table 1.

Table 1: Setting for sensitivity experiments.

|  | Internal grid size | Spacing | Number | Number in one grid |
|---|---|---|---|---|
| Grid experiment | $5D$ | $5D$ | 25600 | 1*1 |
|  | $10D$ | $5D$ | 25600 | 2*2 |
|  | $15D$ | $5D$ | 25600 | 3*3 |
|  | $20D$ | $5D$ | 25600 | 4*4 |
|  | $25D$ | $5D$ | 25600 | 5*5 |
| Space experiment | $10D$ | $2D$ | 25600 | 5*5 |
|  | $15D$ | $3D$ | 25600 | 5*5 |
|  | $20D$ | $4D$ | 25600 | 5*5 |
|  | $25D$ | $5D$ | 25600 | 5*5 |


In the Grid experiment, the size of the inner grid size is changed with an interval of 5D, so that the
number of turbines in one grid varies. As for the first experiment, the inner grid size is 5D, equaling to
the turbine spacing, so there is only one turbine in one grid and has no subgrid wake effect in the original
and new parameterization schemes.   With the number of turbines in the grid changing, it is easily to
observe the sensitivity of the subgrid wake effect to the grid size. In the Space experiment, the scale of
the wind farm is kept unchanged and only the turbine spacing is adjusted. In order to keep the same
number of grid turbines, i.e., to ensure the consistency of the subgrid wake effect on the number
superposition, the simulated grid size is changed to adapt to the change of turbine spacing.The simulation
results of the two groups of experiments are processed and shown in Figure 15 and 16.



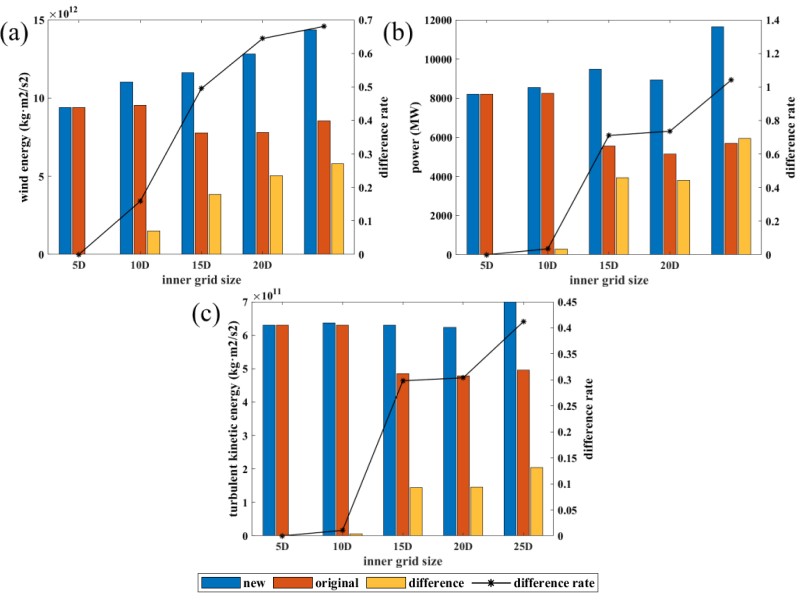


Figure 14: Comparison of simulation results between the original and new schemes for the Grid experiments for (a) the wind energy; (b) the power output; (c) the turbulent kinetic energy. The difference rate is calculated as the difference between results of two schemes being divided by the old scheme results.

As shown in Fig. 14, the simulation results indicate that compared with the inner grid size of 5D, when there is no subgrid wake effect, the wind energy, the output power and the turbulent kinetic energy are all significantly reduced with the grid size, which confirms the influence of subgrid wake effect on the simulation. In these Grid experiments, the size of the grid is gradually enlarged, while the spacing and the total number of turbines remain unchanged, so the range of the wind farm remains unchanged. Theoretically, the simulation results of the wind farm should also remain unchanged, however, the simulation varies due to the various intensity of the subgrid wake effect associated with different grid size. It can be found that the difference rate increases gradually with the increase of grid size. This is because the larger the grid size, the more turbines contained in the same grid, correspondingly, the more significant the subgrid wake effect in the original parameterization scheme. Under the same conditions, when the mesh size is larger, the subgrid wake effect is more significant, and it is more necessary to employ the new wind farm parameterization scheme. As the number of turbines in the grid diminishes, the difference between the original and new parameterization schemes is gradually reduced. When there is only one turbine in the grid, i.e., there is no subgrid wake effect, results of the original and new parameterization schemes are the same, proving that the new parameterization scheme can be compatible





with the original parameterization scheme.
For the Space experiments (Fig. 15), the number of turbines in one grid remains unchanged, while the
spacing of turbines is gradually shortened. The smaller the turbine spacing is, the stronger the wake effect
would be on the downstream turbines. One can see that the difference between the new and original
parameterization schemes for the parameter result is gradually growing, which confirms the enhancement
of the subgrid wake effect. The total amount of wind energy in the wind farm area rises with the turbine
spacing, because the larger area of the wind farm leads to the greater total amount of wind energy, also
it is the case for the total output power and turbulent kinetic energy. In general, this set of experiments
can show that the subgrid wake effect becomes more significant when the turbine spacing is smaller
when the new parameterization scheme should be adopted.
Through the sensitivity experiments the following conclusions can be drawn: the larger the grid size and
the smaller the turbine spacing, the more significant the subgrid wake effect is, and the more suitable to
adopt the new wind farm parameterization scheme which considers the subgrid wake effect.

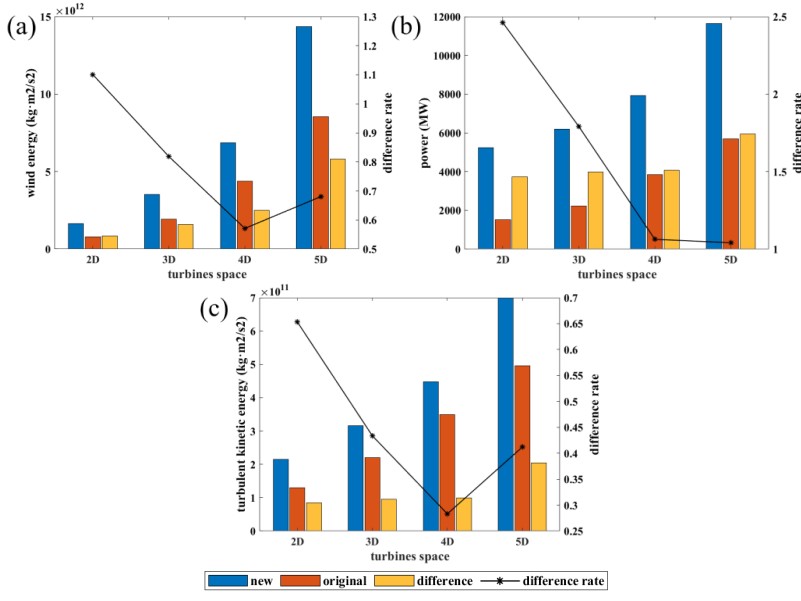


432                        Figure 15: The same as Figure 14 but for the Space experiments.

**5 Summary and discussion**
Based on the engineering wake model of wind turbines, a wake superposition coefficient and an angle
correction coefficient are proposed to be included in the original Fitch scheme to form a new





parameterization scheme of wind farms. The accuracy of the engineering model is improved using the
CFD simulation of the turbine wake. The verification and sensitivity analysis of the new parameterization
scheme are carried out.
In the existing Fitch scheme of WRF, the inflow wind speed of all turbines in one grid are the same,
which ignores the wake effect between turbines. The wake superposition coefficient corrects the subgrid
wake effect under the condition of 0 ° inflow wind angle, and the angle correction coefficient further
corrects the condition under any inflow wind angle. An engineering wake model is calibrated and
modified based on the CFD simulation results. The wake expansion coefficient in the wake analytical
model is calibrated by the change of wake radius of single turbine under different inflow wind speed
conditions. At the same time, the velocity calibration of the wake superposition model is carried out by
the wake superposition of two turbines under different inflow wind speed conditions. The above
correction coefficients are applied to WRF to present a new parameterization scheme of wind farms.
Verification and sensitivity experiments of the new parameterization scheme are carried out, compared
with the original parameterization scheme under different simulation conditions. The experimental
results show that the simulation results of wind energy, power output and turbulent kinetic energy of the
new parameterization scheme are significantly higher than those of the original scheme. The differences
between them are analyzed to be caused by the overestimation of the wind energy absorbed by the
turbines in the grid in the original scheme. Sensitivity experiments show that in the experimental grid
size range (5D~25D), with the increase of grid size, the difference rate between the original and new
schemes grows gradually. In experiments of different turbine spacing (2D~5D), with the shortened
turbine spacing, the different rate between the new and the original schemes is increased gradually. Due
to the limitations engineering practice, there are still some shortcomings in the improved scheme. The
method of solving the angle correction coefficient should be optimized. In the process of solving the
angle correction coefficient, an average method is used to deal with the inflow angle, which cannot
accurately represent the inflow wind speed in front of a specific turbine to a certain extent. It is hoped
that other solving methods can be explored in future to compare the differences with the solution in this
paper and improve the accuracy of the solution. The new and original parameterization schemes should
be used to systematically carry out the experiments of wind farm's influence on various weather and
climate systems, so as to investigate the application performance of the new parameterization schemes
more systematically.



In future, after the new parameterization scheme is verified systematically, it is hoped that the new
scheme can be promoted and integrated into the WRF parameterization scheme for wind farms, so as to
make the simulation of wind farms in WRF more accurate, and provide a better tool to estimate the wind
power and study the environmental impact of wind farms.

*Code availability.* The source code modifying the wind farm module in the WRF can be accessed
through this link (https://doi.org/10.5281/zenodo.8253825).

*Author contributions.* Methodology and experiment: WL, SC, JX, SD, PY. Funding acquisition: JX,
SC. Original draft preparation: WL, SC. Manuscript review and editing: SC, JX, XY, PY.

*Competing interests.* The authors declare that they have no conflict of interest.

**Acknowledgements**
This study is supported by funds from Shenzhen Science and Technology Innovation Committee
(WDZC20200819105831001), the Guangdong Basic and Applied Basic Research Foundation
(2022B1515130006). SC is also supported by the Scientific Research Start-up Fund (QD2021021C).

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
