# Peer review of "Inclusion of the subgrid wake effect between turbines in the wind farm parameterization of WRF"

_Geoscientific Model Development, 2023_

## Referee Comment (RC1)

**Review of "Inclusion of the subgrid wake effect between turbines in the wind farm parameterization of WRF" by Liu, Yang, Chen, Deng, Yu, and Xing, submitted to Geoscientific Model Development**

The authors describe a modification to the default Fitch wind farm parameterization (WFP hereafter) in the WRF model to incorporate the sub-grid wakes between turbines that are located in the same grid cell. The WFP is based on the Jensen linear expansion wake model and accounts for (or claims to account for) the wake overlapping and for the angle between the wind direction and the rows and columns of turbines in a wind farm with a regular layout. The WRF model is then run for 3 days with an incredibly huge wind farm consisting of 25600 turbines of 3 MW each (corresponding to an astronomical installed capacity of 76 GW), covering over 100 x 100 km² offshore of Hong Kong. Sensitivity to grid spacing and turbine spacing is assessed. Results indicate that the proposed modification to the Fitch scheme produces more wind power and more turbulence than the original Fitch scheme, and therefore, somehow, this means that the proposed modification is desirable. Since there is no comparison of the two against any real data, one cannot conclude that the proposed changes are beneficial in any way. This is a fundamental flaw of the paper. In addition, there are several major issues in the way the parameterization was designed, as described below, and the notation is confusing or possibly wrong. The issues are so severe that I cannot recommend publication of the paper at this point.

**Major Remarks**

1. If a team wants to demonstrate that their method is better than the default method, then they need to have an observable dataset to compare the results obtained with the old and the new methods. Only by comparison with observations one can possibly conclude that one is better than the other. It is not sufficient to simulate a case with both, as the authors did in this study, because we do not know what the truth is. To make the matter worse, why simulate an impossible farm, one that has more turbines than any built before (25,600) and that has a capacity at least 30 times larger than the largest wind farm existing today and covering an area of approximately 100 km × 100 km? There is no way to verify any of your findings or confirm that your modification works properly (and I think it does not, see comment 13);

2. The literature review in the Introduction is limited and old, focusing mostly on studies pre-2014. Many more papers have been published since then, including several with the Fitch parameterization. Two relevant studies in particular have been missed and need to be described here because they introduced analytical wake models into WRF to account for the sub-grid wake effects, similar to what this paper is trying to do: Ma et al. (Wind Energy Science, 2022) about the Jensen wake model and Ma et al. (Wind Energy, 2022) about the Gaussian model, Geometric model, among others, and their ensemble.

3. The discussion of Pan and Archer (2018) is incorrect. They used LES results to calibrate the geometric model, which is an analytical model for the wind turbine wakes, and then they inserted it in the WRF model. There is no need to run LES to

use it. As far as I know, it was the first published paper that treated sub-grid wakes in a mesoscale model, WRF in fact. Then Ma et al. (2022a,b) also incorporated sub-grid models in WRF. As such, this paper is not really introducing "a new way, namely, through a simple engineering wake model" because this has actually already been done and documented. The literature review therefore needs to be slightly rewritten to give proper recognition to the three past studies mentioned above: Pan and Archer (2018) and Ma et al. (2022a,b).

4. Also, there was a code bug in WRF that affected the results of the Fitch parameterization, documented by Archer et al. (MWR, 2020). Which version of the WRF was used here? If between 3.6 and 4.2, then the simulations need to be redone to fix the bug.

5. Eqs. 1-3 are incorrect. They are the same as in the original paper by Fitch et al. (2012), but this is not how the scheme is actually implemented in the WRF model. Archer et al. (2020) mentioned above shows the correct version of the equations as they are implemented in WRF since v3.6. Noticeably, the thrust and power coefficients are a function of hub-height wind speed only, not of $V_{ijk}$ (there is only one $C_p$ and one $C_t$ for the turbine, not one for each vertical level $k$). Also, it is not the wind vector squared that is used for the wind speed tendency. Lastly, $A_{ijk}$ is not the swept area, it is the portion of the swept area that is in vertical level $k$.

6. I am not 100% sure, but the correction coefficients seem to be poorly designed, because they rely on the assumption that the layout of the wind farm is a regular grid with rows and columns not only perpendicular to each other, but also oriented north-south and west-east, respectively. I reached this conclusion because the wind direction of $0^o$ is from the North and $\theta$ is clockwise from the North, following the meteorological convention. Thus Eq. 8 only works if the wind rows are oriented from north to south -and- if the wind is from the North, thus all turbines are fully waked (except the front row of course). In all other cases, there is a partial superposition. The angle correction and the $\gamma$ terms take into accounts cases when the wind direction is not from the north and therefore only a fraction of the area overlaps. However, it seems to me that the layout of the wind farm is still assumed to be regular, with rows and columns at $90^o$ from each other. This is not a reasonable assumption. Modern farm may have variable spacing and non-symmetric and non-regular layouts (e.g., Anholt). If true, this a major flaw of the study such that it should not be published because it uses an underlying assumption that is unrealistic and too restrictive.

7. Eq. 10 does not make sense. Previously, $n$ was the total number of upstream turbines, now, I guess, it is the index of the upstream turbines, previously $j$? Or perhaps $n$ is the index of the last upstream turbine? Also, the thrust coefficient is just a function of wind speed, thus it should not depend on the angle $\theta$ or on the correction coefficient. I do not understand how $v_{n0}$ and $v_{n\theta}$ are calculated. Perhaps with Eq. 14? See below.

8. Eq. 14 does not make sense. How can the velocity vector be a tensor?

9. Eq. 15 does not make sense. What is $v(i,i)$? It might help if you could draw some of the $l(i,j)$ in Figure 3. Which cell is used to determine the upstream flow, basically $u_0$ from Eq. 8? I believe that this is key. There is not only one value of $u_0$ in large farms because many grid cells are upstream. Which value(s) is (are) picked here?

10. Eqs. 17, 18, 19, and 20 not explained or derived. The partial superposition with the Jensen model has been solved before, see for example the review paper by Archer et al. (Applied Energy, 2018), Eq. 1-4. Is there a difference between $R$ and $R(i,j)$? Why now are the indices in parenthesis, as opposed to subscripts like earlier in the paper?

11. Not enough details are provided about the CFD simulations. What CFD simulator was used? How were the turbines arranged? I think north-south and east-west perpendicular, but I am not 100% sure, see comment 6 above. What is the diameter $D$? What is the turbine model, was an actuator disk or line used? How many time steps? What was the grid like? Resolution? Run duration? The one figure shown (Fig. 6) appears to be an instantaneous snapshot, not a time-average as we would need in order to tune the parameterization.

12. The discussion at L. 305-309 seems to be related to the incorrect dependency of $C_t$ on the wind speed at each vertical level $k$ mentioned at 5, which is NOT how Fitch works in the recent versions of WRF. This confirms to me that either the wrong equations have been written here or a very old version of WRF has been used, perhaps one with the code bug.

13. The results do not make sense and suggest that something is wrong in the parameterization.

    - Fig. 12 shows the average wind power generated by the farm after 3 days. In the presence northeasterly flow, as the case here (Figure 10), the boundary grid cells along the northern (top) and the eastern (right) sides experience the strongest wind, especially the ones along the right boundary. The turbines in the grid cells along the northeastern boundary all produce maximum power with the default Fitch scheme because there is no sub-grid wake effect in Fitch. By contrast, with the proposed modification, some of the turbines in this "ribbon" experience losses. Therefore I expect the Fitch run to produce more power along the northeastern edge. Instead, the exact opposite happens! Also, I would expect to see large differences in the middle and towards the outer edge of the farm (southwestern end), because the inclusion of wake effects should create more differences further into the farm. Instead, the two simulations are basically identical except for the northeastern band, where the modified Fitch scheme somehow produces more than the original for the most upstream rows! In addition, the new scheme generates an astonishing double power than the original scheme. This is highly suspicious.

    - The total power generated is very small in both runs. With an installed capacity of 3 MW × 25,600 = 76,800 MW, a production of 11,639 MW and 5703 MW for the modified and original scheme, respectively, correspond to a capacity factor of only 15% and 7%. Why is it so low?

    - To make things worse, the energy left in the farm (Figure 11), a weird concept per se because we would need to know over how many levels and how many hours it was calculated, is larger for the simulation that extracted more power, i.e., the one with the new scheme. I find it hard to conceptualize how higher power extraction also leaves more power behind, everything else being equal. Perhaps the authors need to look over a larger domain than just the wind farm area?

I stopped reading at the Sensitivity experiments due to excessive prior issues.

**Minor Remarks**

14. L. 12: Please replace "explosive" with a more appropriate adjective.

15. L. 14: There should be no citations in the abstract, remove the citation Fitch et al. (2012).

16. L. 15: The sentence does not read well, rephrase to something like: "shortfalls, e.g., it does not consider the wakes behind wind turbines inside the same grid cell."

17. L. 22: You need to spell out "CFD".

18. L. 31-34: I know this is probably a translation issue, but the first three sentences have a lot of words but really do not say much ... consider removing them or combining them into one, concise, and clear sentence that goes straight to the point.

19. L. 41 (and other parts): Replace "decay" with "deficit" throughout the article, which is the term I have seen used in the literature.

20. L. 41: The sentence does not read well, rephrase to something like: "wind speed deficit in offshore farms could reach 16% and their wakes could extend downstream as far as 60 km."

21. L. 44-47: The findings by Baydia Roy and Traiteur (2010) have been shown to be unphysical and not due to the wind farm in the study by Archer et al. (2019, Journal of Turbulence, `https://www.tandfonline.com/doi/full/10.1080/14685248.2019.1572161`), see their Figure 1. Thus, Baydia Roy and Traiteur (2010) should not be cited as a reference for the impacts of wind farms.

22. L. 50-54: Similarly to Baydia Roy and Traiteur (2010), parameterizing a wind farm as a surface roughness element, which is what Barrie and Kirk-Davidoff (2010) did, has been shown to be unphysical by several studies, including the study by Fitch et al. (JClimate, 2013), and Jacobson and Archer (PNAS, 2012). Thus, Barrie and Kirk-Davidoff (2010) should not be cited as a reference for the impacts of wind farms.

23. L. 49: Another study that found modest to negligible impacts of offshore wind farms on precipitation is that by Al Fahel and Archer (BAST, 2020).

24. L. 62: The correct reference for the Fitch scheme is Fitch et al. (2012), published in Monthly Weather Review, but the list of references does not include it. Fitch et al. (2013) is used here, but Fitch et al. (2013) is an application of the Fitch scheme, was published in Journal of Climate, and is not the correct reference here. You need to add Fitch et al. (2012) to the References and cite it here.

25. L. 102-104: the sentence needs to be rewritten.

26. Eq. 1-3: the notation needs to be consistent here and in the rest of the paper. First, $i, j, k$ is a subscript for all terms, except for $N_T^{ijk}$. For consistency, call it $N_{ijk}^T$ or just $N_{ijk}$. Second, later you use $i, j$ as the indices for the turbines in a grid cell (confusingly, by the way, as sometimes $i$ is the upstream turbine, sometimes $j$ is). To address this, I recommend removing $i, j$ from these terms here, you can just state in the text that the equations are valid at each grid cell $i, j$.

27. Eq. 7: comparing Eq. 5 and Eq. 7, it appears that $i$ is the turbine of interest and $j$ is one of the $n$ turbines upstream (also from the summation index in Eq. 9). Thus I would call this $u_{ij}$ (not $u_{ji}$), the wind speed at $i$ caused by turbine $j$, and I would replace $x_j$ with $x_{ij}$, the along-wind distance between $i$ and $j$. Later, L. 218, you define $\gamma_{ij}$ as the shielding factor from turbine $i$ to $j$, the exact opposite convention.

28. Figure 9: What value is used to normalize? $u_0$? Or inflow $u_i$?

29. For Figs. 11-13, I would not use a smooth shaded contour plot, but rather a gridded plot, to see better the individual grid cells. Also, using a color like a greenish blue for the near-zero values makes it hard to tell where the difference is positive and where negative. Use white for near-zero instead. Last, use a finite number of color levels, 8-9, and choose the levels in a smart way (e.g., we do not want a plot to be all blue except for with a few grid cells that are other colors, we want to see a balance of cells with the various colors).

---

## Author Comment (AC1)

**Responses to Reviewers' Comments**

*on the manuscript entitled ' Inclusion of the subgrid wake effect between turbines in the wind farm parameterization of WRF ' submitted to Geoscientific Model Development .*

We sincerely thank the editor and two reviewers for their very constructive comments and suggestions, which helped us to improve our manuscript significantly. We made necessary modifications to address all of concerns raised by two reviewers in the revised manuscript where the changes are highlighted. Major modifications of our manuscript include:

1. The CFD experimental part has been expanded to include validation of single turbine wake simulation results against measured results and introduction of double turbine wake experiments (Section 3).
2. We have updated the version of the WRF to 4.3 and repeated the previous experiments (Section 4).
3. We have updated the organization of the manuscript and rewrite some sections.

**Responses to Reviewer 1:**

The authors describe a modification to the default Fitch wind farm parameterization (WFP hereafter) in the WRF model to incorporate the sub-grid wakes between turbines that are located in the same grid cell. The WFP is based on the Jensen linear expansion wake model and accounts for (or claims to account for) the wake overlapping and for the angle between the wind direction and the rows and columns of turbines in a wind farm with a regular layout. The WRF model is then run for 3 days with an incredibly huge wind farm consisting of 25600 turbines of 3 MW each (corresponding to an astronomical installed capacity of 76 GW), covering over 100 x 100 km2 offshore of Hong Kong. Sensitivity to grid spacing and turbine spacing is assessed. Results indicate that the proposed modification to the Fitch scheme produces more wind power and more turbulence than the original Fitch scheme, and therefore, somehow, this means that the proposed modification is desirable. Since there is no comparison of the two against any real data, one cannot conclude that the proposed changes are beneficial in any way. This is a fundamental flaw of the paper. In addition, there are several major issues in the way the parameterization was designed, as described below, and the notation is confusing or possibly wrong. The issues are so severe that I cannot recommend publication of the paper at this point.

Thanks a lot for your such careful and detailed comments. We have made a great effort to improve this manuscript under your suggestions. Hopefully most of your concern has been resolved now.

1. If a team wants to demonstrate that their method is better than the default method, then they need to have an observable dataset to compare the results obtained with the old and the new methods. Only by comparison with observations one can possibly conclude that one is better than the other. It is not sufficient to simulate a case with both, as the authors did in this study, because we do not know what the truth is. To make the matter worse, why simulate an impossible farm, one that has more turbines than any built before (25,600) and that has a capacity at least 30 times larger than the largest wind farm existing today and covering an area of approximately 100 km × 100 km?

There is no way to verify any of your findings or confirm that your modification works properly (and I think it does not, see comment 13);

As the focus of this paper is a large-scale wind farm that is yet to be built, comparing and validating measured data from existing wind farms is not feasible. Although this paper does not utilize the wind farm's measured data to directly validate the new parameterization, sensitivity experiments show that the difference between the original and new parameterization diminishes with the increasing turbine distance, demonstrating a reasonable improvement of the new parameterization scheme. And the CFD experiments are validated by the measured data close to the wind turbine to prove their accuracy. The modified analytical model of the wake flow is validated and corrected by the CFD experiments.

2. The literature review in the Introduction is limited and old, focusing mostly on studies pre-2014. Many more papers have been published since then, including several with the Fitch parameterization. Two relevant studies in particular have been missed and need to be described here because they introduced analytical wake models into WRF to account for the sub-grid wake effects, similar to what this paper is trying to do: Ma et al. (Wind Energy Science, 2022) about the Jensen wake model and Ma et al. (Wind Energy, 2022) about the Gaussian model, Geometric model, among others, and their ensemble.

These references you mention have been added and discussed in the introduction. There are currently four ways to correct the subgrid wake effect between turbines in the WFP of WRF. Pan and Archer et al's (2018) hybrid parameterization scheme based on geometric computation, Ma and Archer's Jensen wake model correction scheme, Xie and Archer's (2020) Gaussian wake model correction scheme, and an ensemble correction scheme that averages the first three. Ma conducted a study and found that the correction schemes presented above performed well, with an overall bias ranging from -11.0% to 21.6% and an overall RMSE between 5.3% and 27.1%. Nonetheless, no individual correction scheme outperformed the others for all wind directions, wind farms, and resolutions.

3. The discussion of Pan and Archer (2018) is incorrect. They used LES results to calibrate the geometric model, which is an analytical model for the wind turbine wakes, and then they inserted it in the WRF model. There is no need to run LES to use it. As far as I know, it was the first published paper that treated sub-grid wakes in a mesoscale model, WRF in fact. Then Ma et al. (2022a,b) also incorporated sub-grid models in WRF. As such, this paper is not really introducing "a new way, namely, through a simple engineering wake model" because this has actually already been done and documented. The literature review therefore needs to be slightly rewritten to give proper recognition to the three past studies mentioned above: Pan and Archer (2018) and Ma et al. (2022a,b).

Thank you very much for your comments. We have cited the researchers mentioned and restructured the section accordingly.

Here is the revised version of the text.

"Pan and Archer et al. (2018) combined the simulation results of LES with the relevant geometric parameters of wind farm layout, and proposed a "hybrid parameterization" scheme. The geometric model was calibrated using the LES results and subsequently integrated into the WRF model and experimental results show that the hybrid parameterization scheme also has a good effect on the correction of subgrid wake effect.

Ma et al. (2022a,b) also incorporated sub-grid models in WRF. They carried out comparative experiments on four schemes: a hybrid parameterization scheme for geometric calculations, a Jensen wake model correction scheme, a Gaussian wake model correction scheme, and an ensemble correction scheme that combines the first three using averaging. All of the correction schemes presented above performed well, but no individual correction scheme outperformed the others for all wind directions, wind farms, and resolutions."

We present a hybrid parameterization scheme based on geometric computation, proposed by Pan and Archer et al. (2018), combined with the Jensen wake model correction scheme proposed by Ma and Archer et al. (2022). Meanwhile, the Ma and Archer et al. (2022) proposal for the Jensen wake model correction scheme only includes wind turbines with an upwind direction of ±30° and distance less than 20D or less in the wake superposition, whereas this paper's analysis does not impose such limitations.

4. Also, there was a code bug in WRF that affected the results of the Fitch parameterization, documented by Archer et al. (MWR, 2020). Which version of the WRF was used here? If between 3.6 and 4.2, then the simulations need to be redone to fifix the bug.

Thank you so much for your suggestion. Regrettably, our simulation runs on version 3.9 of WRF. To rule out the possibility of bugs by the old WRF version, we have updated the version of the WRF to 4.3 and repeated the previous experiments, and we now have significantly more reasonable results than before.

5. Eqs. 1-3 are incorrect. They are the same as in the original paper by Fitch et al. (2012), but this is not how the scheme is actually implemented in the WRF model. Archer et al. (2020) mentioned above shows the correct version of the equations as they are implemented in WRF since v3.6. Noticeably, the thrust and power coefficients are a function of hub-height wind speed only, not of $V_{ijk}$ (there is only one $C_p$ and one $C_t$ for the turbine, not one for each vertical level $k$). Also, it is not the wind vector squared that is used for the wind speed tendency. Lastly, $A_{ijk}$ is not the swept area, it is the portion of the swept area that is in vertical level $k$.

Equations (1) to (3) have been replaced with:

To obtain the vertical distribution of TKE and velocity, the basic principle is that each vertical level $k$ that intersects the rotor contributes proportionally to the fractional rotor area contained in that level $A_k$ and to the horizontal wind speed at that level $U_k$:

$$\frac{\partial \mathrm{TKE}_k}{\partial t} = \frac{1}{2}\frac{A_k C_{\mathrm{TKE}} U_k^3}{(z_{k+1} - z_k)}, \tag{3}$$

$$\frac{\partial u_k}{\partial t} = -\frac{1}{2}\frac{A_k C_T U_k u_k}{(z_{k+1} - z_k)}, \quad \text{and} \tag{4}$$

$$\frac{\partial v_k}{\partial t} = -\frac{1}{2}\frac{A_k C_T U_k v_k}{(z_{k+1} - z_k)}, \tag{5}$$

where $u_k$ and $v_k$ are the horizontal wind components and $z_k$ is the height of vertical level $k$. Equations (3)–(5) are multiplied by a correction factor if energy conservation is not met across the rotor. If multiple turbines are present in the same grid cell, each will add the exact same contribution to the TKE and momentum tendencies as in Eq. (3)–(5).

6. I am not 100% sure, but the correction coefficients seem to be poorly designed, because they rely on the assumption that the layout of the wind farm is a regular grid with rows and columns not only perpendicular to each other, but also oriented north-south and west-east, respectively. I reached this conclusion because the wind direction of $0°$ is from the North and $\theta$ is clockwise from the North, following the meteorological convention. Thus Eq. 8 only works if the wind rows are oriented from north to south - and- if the wind is from the North, thus all turbines are fully waked (except the front row of course). In all other cases, there is a partial superposition. The angle correction and the $\gamma$ terms take into accounts cases when the wind direction is not from the north and therefore only a fraction of the area overlaps. However, it seems to me that the layout of the wind farm is still assumed to be regular, with rows and columns at $90°$ from each other. This is not a reasonable assumption. Modern farm may have variable spacing and non-symmetric and non-regular layouts (e.g., Anholt). If true, this a major flaw of the study such that it should not be published because it uses an underlying assumption that is unrealistic and too restrictive.

The wake shading coefficient's solution process assumes that the wind farm is orientated parallel to the latitude and longitude. The wind farm's vast size has resulted in many grids within the wind farm area. This assumption only affects the grids on the periphery and has no bearing on the inner grids, thus the error of this assumption is limited for a large wind farm where the sub-grid wake effect is notable. And one could narrow the grid size to avoid the inconvenience associated with this assumption. More importantly, the angular correction coefficient and the expansion correction coefficient have been calculated in this paper using CFD experiments. This ensures that the incoming wind speed of each turbine can be accurately calculated, even when the wind farms are not arranged in an east-west or north-south direction. The calculations are based on the wake of the front turbine combined with superposition effect of the wake, rather than on the layout of the wind farm. Therefore, we believe that the new parameterization scheme can still work relatively well in modern wind farm layout schemes.

7. Eq. 10 does not make sense. Previously, $n$ was the total number of upstream turbines, now, I guess, it is the index of the upstream turbines, previously $j$? Or perhaps $n$ is the index of the last upstream turbine? Also, the thrust coefficient is just a function of wind speed, thus it should not depend on the angle $\theta$ or on the correction coefficient. I do not understand how $v_{n0}$ and $v_{n\theta}$ are calculated. Perhaps with Eq. 14? See below.

The subscript 'n' in this context refers to the number of the turbine, rather than the total number of turbines. The reason for not using the subscript 'j' is to avoid confusion, as the subscripts 'i' and 'j' are already used to indicate the horizontal position of the grid. vn0 and vnθ are computed using the geometric shading-based wind farm model specified further down.

8. Eq. 14 does not make sense. How can the velocity vector be a tensor?

v(i,j) refers to the magnitude of wind speed of turbine i at turbine j and is not a vector. We apologize for any confusion caused by the expression. However, **V** in this context does not represent a tensor. Instead, it denotes the influence of the front wind turbine on each other turbine in the unit grid (Eq. 5).

9. Eq. 15 does not make sense. What is $v(i, i)$? It might help if you could draw some of the $l(i, j)$ in Figure 3. Which cell is used to determine the upstream flow, basically $u_0$

from Eq. 8? I believe that this is key. There is not only one value of $u_0$ in large farms because many grid cells are upstream. Which value(s) is (are) picked here?

The incoming wind speed of each turbine, $v(i,i)$, is determined after the preceding turbine effects are taken into account. The wind speed of the current grid, $u_0$, varies depending on the location of the grid.

10. Eqs. 17, 18, 19, and 20 not explained or derived. The partial superposition with the Jensen model has been solved before, see for example the review paper by Archer et al. (Applied Energy, 2018), Eq. 1-4. Is there a difference between R and R(i, j)? Why now are the indices in parenthesis, as opposed to subscripts like earlier in the paper?

R refers to the radius of the turbine, while $R(i, j)$ refers to the radius of turbine i's wake at turbine j. The two meanings are distinct. However, there appears to be a deficiency in the work carried out as we have not achieved consistency with the prior velocity representation in relation to the shading radius (according to the comment from reviewer 2). Currently, $i$ has been harmonized as the turbine of interest while $j$ is the turbine that produces the effect.

11. Not enough details are provided about the CFD simulations. What CFD simulator was used? How were the turbines arranged? I think north-south and east-west perpendicular, but I am not 100% sure, see comment 6 above. What is the diameter D? What is the turbine model, was an actuator disk or line used? How many time steps? What was the grid like? Resolution? Run duration? The one figure shown (Fig. 6) appears to be an instantaneous snapshot, not a time-average as we would need in order to tune the parameterization.

The details of the CFD experimental section has been added (L. 283-330). We regret that we could not include this part of the work previously. However, this section has undergone considerable effort and has been verified with actual radar wind measurement data. As the RANS method is utilized, the velocities displayed in Figure 6 are representative of the time-averaged concept. These results may differ from those of the time-averaged simulation conducted using the LES method. It is important to note, however, that this only represents a portion of our research. In reality, we computed the RANS outcomes for the blade at six different angles and collated an average to achieve a desirable outcome in the end.

[Figure]

12. The discussion at L. 305-309 seems to be related to the incorrect dependency of Ct on the wind speed at each vertical level k mentioned at 5, which is NOT how Fitch works in the recent versions of WRF. This confirms to me that either the wrong equations have been written here or a very old version of WRF has been used, perhaps one with the code bug.

We have updated the version of the WRF to 4.3 and repeated the previous experiments.

13. The results do not make sense and suggest that something is wrong in the parameterization.

• Fig. 12 shows the average wind power generated by the farm after 3 days. In the presence northeasterly flow, as the case here (Figure 10), the boundary grid cells along the northern (top) and the eastern (right) sides experience the strongest wind, especially the ones along the right boundary. The turbines in the grid cells along the northeastern boundary all produce maximum power with the default Fitch scheme because there is no sub-grid wake effect in Fitch. By contrast, with the proposed modification, some of the turbines in this "ribbon" experience losses. Therefore I expect the Fitch run to produce more power along the northeastern edge. Instead, the exact opposite happens! Also, I would expect to see large differences in the middle and towards the outer edge of the farm (southwestern end), because the inclusion of wake effects should create more differences further into the farm. Instead, the two simulations are basically identical except for the northeastern band, where the modified Fitch scheme somehow produces more than the original for the most upstream rows! In addition, the new scheme generates an astonishing double power than the original scheme. This is highly suspicious.

• The total power generated is very small in both runs. With an installed capacity of 3 MW × 25,600 = 76,800 MW, a production of 11,639 MW and 5703 MW for the modified and original scheme, respectively, correspond to a capacity factor of only 15% and 7%. Why is it so low?

• To make things worse, the energy left in the farm (Figure 11), a weird concept per se because we would need to know over how many levels and how many hours it was calculated, is larger for the simulation that extracted more power, i.e., the one with the new scheme. I find it hard to conceptualize how higher power extraction also leaves more power behind, everything else being equal. Perhaps the authors need to look over a larger domain than just the wind farm area?

• Version 4.3 of the WRF has been utilized to re-run the experiment, and the new outcomes are assessed. The analysis now uses the time-averaged outcomes for the last six hours of the simulation, rather than the snapshot results. From our current results, the total power of the wind farm is now closer to the expected under the original and new parameterization schemes, but the spatial distribution of the power is different. In the back row grid of the wind farm, the total power in the new scheme is lower than in the original scheme due to the added wake effect, which is realistic. However, at the right edge of the wind farm, the power at the edge of the grid is slightly higher than in the original scheme due to the higher incoming wind speed calculated in the new scheme, even though the wake effect is taken into account.

• The low wind power output stems from the wind farm's large size and the subsequent wake impact on internal turbines from turbines at the edge. This lowers the internal turbine output and ultimately affects the total power output. Perhaps if not every neighboring grid in this wind farm had turbines set up, but instead there exists a portion of the grid where no turbines are set up to provide some space for wake recovery, the normalized power of the whole wind farm would be much higher.

• We have already addressed this question in the first point. For reference, we provide results for a more extensive area (Figure 12). It has been discovered that the new parameterization scheme does, in fact, decrease the extraction of wind kinetic energy within the wind farm area. As a result, this effect impacts the downstream portion of the wind farm, leading to an increase in downstream wind kinetic energy. The observed phenomenon is not solely attributable to the decrease in turbine power, as the power shift within the wind farm is negligible in the updated findings. It seems that the downstream alteration is more likely a result of heightened turbulent kinetic energy comparing to the original parameterization. This enhances the entrainment of momentum in the wake and upper atmosphere, in other words, strengthens the turbulent transport process of mean kinetic energy from the ambient region to the wake region.

[Figure]

Figure12 Comparison of wind energy distribution in wind farm area for (a) the new parameterization scheme; (b) the original parameterization scheme; (c) their difference.

Minor Remarks

14. L. 12: Please replace "explosive" with a more appropriate adjective.

Corresponding expressions have been modified. (L. 12-13)

15. L. 14: There should be no citations in the abstract, remove the citation Fitch et al. (2012).

Accepted and revised. (L. 15)

16. L. 15: The sentence does not read well, rephrase to something like: "shortfalls, e.g., it does not consider the wakes behind wind turbines inside the same grid cell."

Many thanks for your suggestion. We have revised the sentence. (L. 16-18)

17. L. 22: You need to spell out "CFD"

Accepted and revised. (L. 22)

18. L. 31-34: I know this is probably a translation issue, but the first three sentences have a lot of words but really do not say much ... consider removing them or combining them into one, concise, and clear sentence that goes straight to the point.

Thank you for your comments. We rewrite this one sentence to make it more concise and to the point. (L. 34-36)

19. L. 41 (and other parts): Replace "decay" with "deficit" throughout the article, which is the term I have seen used in the literature.

Thank you for your suggestion. This is indeed a more appropriate expression and we have made the substitution. (L. 42, 45)

20. L. 41: The sentence does not read well, rephrase to something like: "wind speed deficit in offshore farms could reach 16% and their wakes could extend downstream as far as 60 km."

Thank you for your advice and we have adopted the expression you suggested. (L. 41-42)

21. L. 44-47: The findings by Baydia Roy and Traiteur (2010) have been shown to be unphysical and not due to the wind farm in the study by Archer et al. (2019, Journal of Turbulence, https://www.tandfonline.com/doi/full/10.1080/14685248. 2019.1572161), see their Figure 1. Thus, Baydia Roy and Traiteur (2010) should not be cited as a reference for the impacts of wind farms.

Many thanks for your suggestion. We have removed the citation to their paper. (L. 45)

22. L. 50-54: Similarly to Baydia Roy and Traiteur (2010), parameterizing a wind farm as a surface roughness element, which is what Barrie and Kirk-Davidoff (2010) did, has been shown to be unphysical by several studies, including the study by Fitch et al. (JClimate, 2013), and Jacobson and Archer (PNAS, 2012). Thus, Barrie and Kirk-Davidoffff (2010) should not be cited as a reference for the impacts of wind farms.

Thank you for your suggestion. We have removed the citation to their paper. (L. 48)

23. L. 49: Another study that found modest to negligible impacts of offshore wind farms on precipitation is that by Al Fahel and Archer (BAST, 2020).

Many thanks for your suggestion. This is indeed a paper we do not read. However, our research group has uncovered that offshore wind farms' influence on precipitation is more intricate and can be instigated by topographic effects downstream in the region.

24. L. 62: The correct reference for the Fitch scheme is Fitch et al. (2012), published in Monthly Weather Review, but the list of references does not include it. Fitch et al. (2013) is used here, but Fitch et al. (2013) is an application of the Fitch scheme, was published in Journal of Climate, and is not the correct reference here. You need to add Fitch et al. (2012) to the References and cite it here.

Thank you for your suggestion. We have revised this part. (L. 57)

25. L. 102-104: the sentence needs to be rewritten.

Accepted and revised. (L. 93-95)

26. Eq. 1-3: the notation needs to be consistent here and in the rest of the paper. First, i, j, k is a subscript for all terms, except for $N_T^{ijk}$. For consistency, call it $N_T^{ijk}$ or just $N_{ijk}$. Second, later you use $i, j$ as the indices for the turbines in a grid cell (confusingly, by the way, as sometimes $i$ is the upstream turbine, sometimes $j$ is). To address this, I recommend removing $i, j$ from these terms here, you can just state in the text that the equations are valid at each grid cell $i, j$.

Many thanks for your suggestion. As we mentioned in the fifth reply, we rewrote Eq. 1-3. (L. 103-105)

27. Eq. 7: comparing Eq. 5 and Eq. 7, it appears that i is the turbine of interest and $j$ is one of the n turbines upstream (also from the summation index in Eq. 9). Thus I would call this $u_{ij}$ (not $u_{ji}$), the wind speed at $i$ caused by turbine $j$, and I would replace $x_j$ with $x_{ij}$, the along-wind distance between $i$ and $j$. Later, L. 218, you define $\gamma_{ij}$ as the shielding factor from turbine $i$ to $j$, the exact opposite convention.

Thank you for your suggestion. The corresponding section subscripts have now been adjusted to make them consistent. (L. 216, 219, 220, 227)

28. Figure 9: What value is used to normalize? $u_0$? Or inflow $u_i$?

Wind speed is normalized by $u_0$ and we provide additional explanations in the text. (L. 347)

29. For Figs. 11-13, I would not use a smooth shaded contour plot, but rather a gridded plot, to see better the individual grid cells. Also, using a color like a greenish blue for the near-zero values makes it hard to tell where the difference is positive and where negative. Use white for near-zero instead. Last, use a finite number of color levels, 8-9, and choose the levels in a smart way (e.g., we do not want a plot to be all blue except for with a few grid cells that are other colors, we want to see a balance of cells with the various colors).

Many thanks for your suggestion. We have revised this part and replaced some of the figure. (Figs. 13-15)

---

## Author Comment (AC2)

**Responses to Reviewers' Comments**

*on the manuscript entitled ' Inclusion of the subgrid wake effect between turbines in the wind farm parameterization of WRF ' submitted to Geoscientific Model Development .*

We sincerely thank the editor and two reviewers for their very constructive comments and suggestions, which helped us to improve our manuscript significantly. We made necessary modifications to address all of concerns raised by two reviewers in the revised manuscript where the changes are highlighted. Major modifications of our manuscript include:

1. The CFD experimental part has been expanded to include validation of single turbine wake simulation results against measured results and introduction of double turbine wake experiments (Section 3).
2. We have updated the version of the WRF to 4.3 and repeated the previous experiments (Section 4).
3. We have updated the organization of the manuscript and rewrite some sections.

**Responses to Reviewer 2:**

Here the authors try to include subgrid wake effects in the Fitch scheme, which is the WRF's default wind farm parametrization. The manuscript is at this state and stage very weak both in its conceptualization, its presentation, and its results, and therefore, I cannot suggest its publication. I became very worried about this work, so I stopped reading it after I saw the authors even carried out sensitivity experiments as well! So, my review goes until Section 4.1. I apologize but my opinion is that this manuscript needs major surgery before it can be submitted again to a proper journal as GMD. Unfortunately, I have to recommend its rejection, but I encourage the authors to revise it thoroughly. Also, I strongly recommend sending this paper to a English grammar office since the language can be also largely improved; it is not much about misspellings but about the way to write science per se.

Thank you for your comments and we have revised the article based on your request. We have identified issues with the previously presented effects based on the feedback you provided and we have now obtained more accurate and reliable results.

1. CFD: already in the abstract you mentioned that the coefficients will be derived from CFD results, but you do not say anything about what do you mean by CFD here. Moreover, you do not mention what CFD tool you used later in the manuscript. CFD can be anything; it can even be WRF itself! Since it is an important part of your study, the "CFD" should be properly described, is it LES, RANS, uRANS? What is the turbine model in it? How is turbulence model? Etc.

Thank you very much for your comments. The details of the CFD experimental section have been added (L. 283-330). We use the RANS method with a solid model for the turbine and a realizable k-$\varepsilon$ model for the turbulence model.

2. Section 3.1: As mentioned in point 1 there is no description of the so-called CFD model. But also important is that from the text, it sounds as if the subgrid model you

end up using is based on Eq. (4), which is determined by the coefficients you estimate based on your specific CFD precomputations. This reads as if the layout of the wind farm or the turbine itself is different from that you used in your CFD precomputations, then these coefficients in Eq. (4) will also differ (or are they generic?). Then, this ends up not being a generic wind farm parameterization anymore because (I guess) you did not simulate all possible layouts within a grid cell or all types of turbines.

As far as we know, a significant number of offshore wind farms (in the South China Sea area) are constructed according to a regular layout. This paper focuses on the construction of future mega wind farms, which also tend to have a regular pattern within the majority of their occupied grids. Although certain variations may exist at the wind farm edges, their impact on the entire facility is small. More importantly, the angular correction coefficient and the expansion correction coefficient have been calculated in this paper using CFD experiments. This ensures that the incoming wind speed of each turbine can be accurately calculated, even when the wind farms are not arranged in an east-west or north-south direction. The calculations are based on the wake of the front turbine combined with superposition effect of the wake, rather than on the layout of the wind farm. Therefore, we believe that the new parameterization scheme can still work relatively well in other possible wind farm layout schemes as well as other types of turbines.

3. In order to understand the results shown in Fig. 11, information about the wind direction within the 3 days of simulation is needed. From Fig. 10 we only have a snapshot and it hints as the northeasterly sector is the one dominant but direction can change quickly during these three days.

Thank you so much for your suggestion. Version 4.3 of the WRF software is utilized to re-run the experiment, and the new outcomes are assessed. The analysis now uses the time-averaged outcomes for the last six hours of the simulation, rather than the snapshot results.

4. Also related to point 3 above: It is difficult to understand what do you mean by wind energy in Figure 11 and the text around it in page 16? Do you mean the amount of energy that you could extract from the wind? If so, then it should not depend on the wind farm parameterization but only on the wind climatology without the wind farm (basically you should have run a simulation without then wind farm). However, you have two different results for the two types of parameterization so I guess this is the energy extracted within the 3 days but then the units should be in MW h/day or similar. The numbers you provide in Page 16 are all very strange. What does $1.44 \times 10^{13}$ mean? Or $8.54 \times 10^{12}$? There is no "absorption of wind energy in the wind farm region" How can you know that the error is reduced (line 353), how could you know it was being overestimated? The text in lines 348-357 is just way too weird.

Thank you for your comments. The kinetic energy of the wind in the wind farm area is the remaining energy in the environment after the conversion of electrical energy by the wind turbines. It is calculated based on the wind speed of the wind farm area obtained from simulation. In the original wind farm parameterization scheme, the incoming wind speeds of the turbines in the grid are equal to the incoming wind speeds of the turbines in the "first row" owing to the lack of consideration of the sub-grid wake, leading to an overestimation of the wind kinetic energy extracted from the wind farm. The simulation findings reveal heightened wind kinetic energy within the wind farm,

implying a rectification of the wind energy over-absorption in the previous parameterization method.

From our latest results, the wind energy inside the wind farm region using the new scheme, which is $5.80737 \times 10^{13}$ kg·m$^2$/s$^2$ in total, is higher than that of the original scheme ($5.29162 \times 10^{13}$ kg·m$^2$/s$^2$), increasing by 9.75%. More details can be seen from line 403-412.

5. Figure 12 and text around: Also a very strange plot and description. I guess you do not mean 70 MW in the colorbar but GW? But most importantly, is this power output at a particular time? I mean is this the instantaneous power output or some kind of average power output within the 3 days. Further, the number is strange as your maximum power output should be 76800 MW but you have numbers of 11639 and 5703 MW, so about 10% of the rater power of this mega hyper wind farm.

Thank you for your points. The color-bar denotes the power output of a specific electrical grid, providing a maximum of 75MW per grid (25*3). The unit of measurement for the color-bar in this case is MW with no doubt. Version 4.3 of the WRF software was utilized to re-run the experiment, and the new outcomes were assessed. The analysis now uses the time-averaged outcomes for the last six hours of the simulation, rather than the instantaneous results at a specific moment in time. Regarding the issue of low power, it is possible that this is due to WRF version issues. In the latest results, the total power of the wind farms reaches 22.37GW, which is about 40% of the maximum power output.

6. And perhaps more importantly: if I understood correctly the new parametrization (Fitch and subgrid model) results in larger energy yield than the original one (Fitch only). If the Fitch scheme is basically the same, the effect of the subgrid model should be to lower the energy yield as you are accounting for the effect of wakes within the grid cell. So I do not really understand why is your new scheme yielding more energy.

Thank you for your comments. Version 4.3 of the WRF software has been utilized to re-run the experiment, and the new outcomes were assessed as mentioned before.

The analysis now uses the time-averaged outcomes for the last six hours of the simulation. From our current time-averaged results, the total power of the wind farm is now closer under the original and new parameterization schemes, but the spatial distribution of the power is quite different. In the back row grid of the wind farm, the total power in the new scheme is lower than in the original scheme due to the added wake effect, which is physically intuitive that the wake will lead to a reduction in power. However, at the right edge of the wind farm, the power at the edge of the grid is slightly higher than in the original scheme due to the higher incoming wind speed calculated in the new scheme, even though the wake effect is taken into account.

Specific comments:

1. Line 20: "These coefficients are added in the WRF"; I guess you mean that they were added in a new implementation of the Fitch scheme, which is coupled within the WRF modelling system.

You are correct, and we have made a more accurate statement in the paper. (Line 23)

2. Line 21: "Sensitivity experiments"; here you need to say of what? Spatial resolution, PBL schemes? What kind of sensitivity?

The sensitivity tests presented involve varying distances between turbines. The purpose is to demonstrate an approximation to the original layout when the spacing is sufficiently large. (Line 24)

3. Line 25: "shows more advantages" compared to what?

The benefit lies in the reduction of the overall power of the turbine in the wake region in new scheme compared to the original. We have also explained this in the paper. (Line 28)

4. Line 35: replace "achieving a rapid development period

Accepted and revised. (Line 36)

5. Line 59: full stop after meters and then start a new sentence with "Numerical"

Accepted and revised. (Line 55)

6. Line 76: delete "technology" after "LES"

Accepted and revised. (Line 73)

7. Line 78: replace "LES simulation of wind farm is processed by the" by "LES is combined with"

Accepted and revised. (Line 75)

8. Lines 80-83: these two sentences referring to the work of Elshafei et al. (2021) has nothing to do with your work

Many thanks for your suggestion. We have removed the citation to their paper. (L. 77)

9. At the end of the introduction, you introduce the different sections of your work but not all of them.

Thank you for your suggestion. We have revised this part. (L. 82)

10. (1)—(3) are wrong as these are not those implemented in the original Fitch scheme

Equations (1) to (3) have been replaced with:

To obtain the vertical distribution of TKE and velocity, the basic principle is that each vertical level $k$ that intersects the rotor contributes proportionally to the fractional rotor area contained in that level $A_k$ and to the horizontal wind speed at that level $U_k$:

$$\frac{\partial \text{TKE}_k}{\partial t} = \frac{1}{2} \frac{A_k C_{\text{TKE}} U_k^3}{(z_{k+1} - z_k)}, \quad (3)$$

$$\frac{\partial u_k}{\partial t} = -\frac{1}{2} \frac{A_k C_T U_k u_k}{(z_{k+1} - z_k)}, \quad \text{and} \quad (4)$$

$$\frac{\partial v_k}{\partial t} = -\frac{1}{2} \frac{A_k C_T U_k v_k}{(z_{k+1} - z_k)}, \quad (5)$$

where $u_k$ and $v_k$ are the horizontal wind components and $z_k$ is the height of vertical level $k$. Equations (3)–(5) are multiplied by a correction factor if energy conservation is not met across the rotor. If multiple turbines are present in the same grid cell, each will add the exact same contribution to the TKE and momentum tendencies as in Eq. (3)–(5).

11. (4) and some others: you use the dot product to represent multiplication sometimes. You should not. This is the dot product between two vectors and a coefficient is not a vector

Accepted and revised. (Line 122 Eq. 4 )

12. Line 148: Replace "which is related to the roughness" by "which can be related to the roughness"

Accepted and revised. (Line 140)

13. Line 160: "Eq. (2.5)" there is no such an equation, so do you mean Eq. (5)?

We're sorry we made such a cheap mistake, and we've revised it. (Line 154)

14. Line 286: "two-turbine wake experiment" which one? Did you introduce it before?

We regret that we could not include this part of the work initially due to space restrictions. We have now added this section to the paper. (Line 298-316)

15. Lines 311 and 312: Do you mean "Implementation"?

Accepted and revised. (Line 365-366)

16. Line 339: what is "the kind of speed"?

I am sorry to have caused a deviation in your understanding of a wind speed condition we expressed, and we have made a more accurate statement in the paper. (Line 392)

17. Lines 339-341: please rephrased. What do you mean by conducive?

I am sorry to have caused a deviation in your understanding. Our aim is to convey the evident experimental phenomena that can be observed in numerical simulations under such a high wind speed conditions. (Line 394-395)

18. Fig 11: Is this for the innermost domain? If so please state this in the caption

This is for the innermost domain (wind farm area) and we have state this in the caption.

19. Fig 12: what are the units of the turbulent kinetic energy?

The unit of the turbulent kinetic energy is the same as the wind kinetic energy.